# Benzene Exposure and MicroRNAs Expression: In Vitro, In Vivo and Human Findings

**DOI:** 10.3390/ijerph20031920

**Published:** 2023-01-20

**Authors:** Paola Mozzoni, Diana Poli, Silvana Pinelli, Sara Tagliaferri, Massimo Corradi, Delia Cavallo, Cinzia Lucia Ursini, Daniela Pigini

**Affiliations:** 1Department of Medicine and Surgery, University of Parma, 43126 Parma, Italy; 2CERT, Center of Excellent Research in Toxicology, University of Parma, 43126 Parma, Italy; 3INAIL Research, Department of Occupational and Environmental Medicine, Epidemiology and Hygiene, Via Fontana Candida, 1, 00078 Monte Porzio Catone, Italy

**Keywords:** benzene, microRNA, in vitro, in vivo, human exposure, biomarkers of exposure and effect

## Abstract

MicroRNAs (miRNAs) are important regulators of gene expression and define part of the epigenetic signature. Their influence on human health is established and interest in them is progressively increasing. Environmental and occupational risk factors affecting human health include chemical agents. Benzene represents a pollutant of concern due to its ubiquity and because it may alter gene expression by epigenetic mechanisms, including miRNA expression changes. This review summarizes recent findings on miRNAs associated with benzene exposure considering in vivo, in vitro and human findings in order to better understand the molecular mechanisms through which benzene induces toxic effects and to evaluate whether selected miRNAs may be used as biomarkers associated with benzene exposure. Original research has been included and the study selection, data extraction and assessments agreed with PRISMA criteria. Both in vitro studies and human results showed a variation in miRNAs’ expression after exposure to benzene. In vivo surveys also exhibited this trend, but they cannot be regarded as conclusive because of their small number. However, this review confirms the potential role of miRNAs as “early warning” signals in the biological response induced by exposure to benzene. The importance of identifying miRNAs’ expression, which, once validated, might work as sentinel molecules to better understand the extent of the exposure to xenobiotics, is clear. The identification of miRNAs as a molecular signature associated with specific exposure would be advantageous for disease prevention and health promotion in the workplace.

## 1. Introduction

Benzene is a ubiquitous environmental pollutant produced by natural (e.g., natural gas, petroleum, combustion processes, etc.) and anthropogenic processes (e.g., industrial and vehicular emissions, chemical processes, cigarette smoke, etc.). [1]. Occupational exposure happens in the petrochemical industry, in manufacturing sectors that require aromatic solvents or glues that contain benzene, and it occurs mainly through inhalation [2,3]. According to the European Union, some uses of benzene are restricted under Annex XVII of REACH (Regulation (EC) 1907/2006 of the European Parliament). In particular, it shall not (a) be used in toys or parts of toys containing more than 5 mg/kg; and (b) be placed on the market or used (as a substance or as a constituent of mixture) in concentrations equal to or greater than 0.1% by weight. However, the last ban does not apply to fuels, in which case the benzene level should not exceed 1%, and in the case of use in industrial processes, in which emissions of benzene in higher quantities than those prescribed by the legislation in force are not allowed [4].

In spite of these efforts to control benzene pollution, it remains one of the most dangerous contaminants in urban air. The maximum desirable value under the Directive 2008/50/EC on air quality in Europe [5] is 5 μg/m^3^, which refers to the annual average concentration in urban outdoors. Regarding occupational exposure, the American Conference of Governmental Industrial Hygienists (ACGIH) recommended a time-weighted average-threshold limit value (TLV-TWA) of 0.5 ppm (1.60 mg/m^3^). The Italian occupational exposure limit (OEL) value reported by Legislative Decree 81/08 [6] (transposition of the Directive 2004/37/CE [7]) is 1 ppm (3.25 mg/m^3^ of air), but the Directive 2022/431 that will enter into force on 5 April 2024 will reduce it to 0.2 ppm (equal to 0.66 mg/m^3^). However, the Committee for Risk Assessment (RAC), which supports the European Commission by giving scientific opinions on OEL, proposed a value of 0.05 ppm to protect workers from leukemia, as well as other adverse health effects. The RAC’s opinion is based on the fact that a threshold based on the indirect genotoxic effects of benzene in workers can be used to derive a new OEL [8]. All of this underlines the international community’s attention and efforts to reduce the presence of benzene in the workplace and in the environment.

Benzene was identified as a Group 1 carcinogen and recognized by the International Agency for Research on Cancer (IARC) as a human carcinogen [9], with evidence supporting its carcinogenicity (leukemia) in occupational and non-occupational settings. Its exposure has been extensively studied and there is epidemiological and toxicological evidence showing that high exposure to benzene can lead to the development of acute myeloid leukemia (AML) and nonlymphocytic leukemia (ANLL), as well as myelodysplastic syndrome (MDS), in humans [10,11]. Furthermore, developmental, reproductive, respiratory, immunological and metabolic effects have been also identified [12], although the mechanisms of its toxicity are not entirely clear. Benzene is primarily absorbed by inhalation if either occupational or non-professional exposure occurs, and it is metabolized largely in the liver through reactions catalyzed by two isoforms of cytochrome P450 (CYP) oxidase, 2E1 and 2B1. It is hence oxidized to benzene oxide (BO), and then converted to trans, trans-muconic acid (t,t-MA) and a small fraction in S-phenylglyoxylic acid (S-PMA) by conjugation with glutathione (GSH) from glutathione-S-transferase (GST), before it is finally excreted in the urine. T,t-MA and S-PMA are considered the most sensitive urinary biomarkers for exposure to benzene [13]. Highly reactive benzene intermediates, such as hydroquinone (HQ) and benzoquinones (1,4-BQ and 1,2-BQ), are considered toxic metabolites of benzene, owing to their ability to react with macromolecules and lead to toxic effects. They can also activate reactive oxygen species (ROS) production and cause DNA damage and mutations [13,14]. Consequently, the carcinogenicity of benzene is therefore linked to its metabolism by the production of metabolites and ROS.

Exposure to benzene can lead not only to genetic mutations, associated with DNA and RNA damage, but also to inappropriate gene expression [15]. Moreover, it was found that epigenetic alterations might also play a role in benzene tumorigenesis [16]. The epigenetic alterations comprise DNA methylation, histone modification and microRNA (miRNA) interference [16]. Several studies have indicated that exposure to benzene induces global DNA hypomethylation in the exposed subject [17], as well as in in vitro studies [18,19], or DNA hypermethylation of the tumor suppressor genes p15 and p16 in benzene poisoning workers [20]. Similarly, histone modifications play a crucial role in transcriptional gene regulation, making genes accessible or inaccessible to transcriptional factors [21]. Yu et al. observed H4, H3, and H3K4 histone modifications in the promoter region of topoisomerase Ⅱα (Topo Ⅱα) in subjects exposed to benzene [22]. 1,2,4-Benzenetriol (BT), a minor benzene metabolite, seems to inhibit hemin-induced erythroid differentiation, where DNA methylation and histone acetylation also played important roles by downregulating erythroid-specific genes [23]. 1,4-BQ, the major leukemogenic metabolite of benzene, irreversibly inhibits the human histone methyltransferase SETD2, resulting in decreased H3K36me3 [24]. Finally, association between benzene and aberrant miRNA expression, which is the topic of this review, was also reported in several studies, indicating their association with benzene-induced hemotoxicity and leukemogenesis.

MiRNAs are single-stranded non-coding RNAs, which play important roles in a variety of cell processes (e.g., cell development and growth, differentiation and apoptosis) [25]. Their role in gene regulation was first identified in *Caenorhabditis elegans* in the early 1990s, where the miRNA (LIN 4) negatively regulated the gene LIN14. Afterwards, they were identified to play a major role in the post-transcriptional regulation of human genes [26]. MiRNAs are usually made of 18–25 nucleotides, are extremely specific in tissues and are highly preserved. MiRNAs, binding complementary messenger RNA (mRNA) sequences in the 3′UTR, inhibit their translation, regulating several physiological processes [27]. MiRNAs’ encoding sequences are located in exons or introns of protein-encoding genes or in intergenic regions. The biogenesis of miRNA begins with the synthesis of a long transcript known as a pri-miRNA (see Figure 1). Pri-miRNAs are transcribed predominantly by RNA polymerase II and retain mRNA features, such as a 5′ cap structure and a 3′poly(A) tail. In the nucleus, pri-miRNA is processed to pre-miRNA by RNase III enzyme Drosha-DGCR8 (DiGeorge syndrome critical region gene 8). The resulting precursor hairpin, the pre-miRNA, is exported from the nucleus to the cytoplasm by Exportin-5–Ran-GTP. Then, the RNase Dicer, in complex with the double-stranded RNA-binding protein TRBP, cleaves the pre-miRNA hairpin to its mature length. The functional strand of the mature miRNA is loaded together with Argonaute (Ago2) proteins into the RNA-induced silencing complex (RISC) where it guides the RISC to silence target mRNAs through mRNA cleavage, translational repression or deadenylation, whereas the passenger strand is degraded [28]. They fine-tune the gene expression in response to various external stimuli, including environmental toxicants [29].

Growing evidence that the expression of miRNAs is affected by several known toxicants suggests an important role of miRNAs in toxicology, which could provide a link between environmental influences and gene expression [32]. MiRNAs’ regulatory networks are complex because a single miRNA can target numerous mRNAs, often in combination with other ones. As miRNAs control protein expression, they may become a worthwhile instrument for predicting an effective drug-target [29]. MiRNA profiling could lead to the discovery of miRNAs as biomarkers of exposure/effect, which might work as sentinel molecules to better predict both efficacy and safety of prevention and/or protection measurements. There are several important features of miRNAs that provide potential advantages compared to more established protein biomarkers [33]. Between these, one is their stability in biofluid samples through various protective interactions, such as with microvesicles, proteins and lipids [34], as well as the fact that they exhibit tissue-specific patterns of expression [35].

The aims of this review are as follows: (a) to understand the potential biological implications of miRNA expression/profile changes induced by benzene exposure in order to better understand the molecular mechanisms through which benzene induces toxic effects, particularly in the bone marrow; (b) to compare the results from in vitro, in vivo and human studies; and (c) to evaluate whether selected miRNAs may be used as biomarkers of exposure and/or effect associated with benzene exposure.

## 2. Materials and Methods

### 2.1. Literature Search Methodology

A systematic search was performed in accordance with the PRISMA guidelines [36,37]. A literature search to identify relevant articles was carried out until 30 August 2022 using the electronic bibliographic databases Pubmed and Scopus. The keywords used for the searching scheme (in the titles or abstracts) were an association of “benzene” and “microRNA” (first key words) and, as a second key words, “exposure”, “biomonitoring”, “human”, “in vitro” and “in vivo”.

### 2.2. Eligibility Criteria and Study Selection

Only studies with original data on benzene exposure and miRNA profile were acceptable for this review. Original papers published in English, Italian or Spanish until 30 August 2022 were included. For more than one article on the same study, the most recent or comprehensive one was chosen. The selection was made by first reading the abstract and then the full article. Studies were excluded in accordance with the following criteria: (1) case reports, meta-analysis, reviews and letters; (2) duplicated data and incomplete information studies; and (3) papers not written in English, Italian or Spanish. Two evaluators carried out electronic searches independently using the inclusion and exclusion criteria. Discrepancies in interpretation and disagreements were resolved by a panel discussion and, if necessary, by consultation with the third author. Figure 2 displays a flowchart of the process by which the articles were selected.

Seven in vitro studies were considered and were selected on the basis of the exposure of different cell lines to increasing concentrations of a typical toxic metabolite of benzene, 1,4-BQ and HQ, the phenolic metabolite of benzene with a potential risk for hematological disorders and hematotoxicity in humans. Regarding studies on animals, only three papers were identified. They were conducted on mice using different routes (inhalation or injection) and timing of exposure (acute to chronic). The selected human exposure studies considered the following: (1) benzene exposure in the workplace; (2) retrospective studies; and (3) exposure to environmental pollution or smoking habits.

All selected articles were compared on the basis of the results and various analytical methods were not taken into consideration. The bias assessment in the studies was performed using the Cochrane bias risk tool [38]. Considering the different types of biases requested by the tool, incomplete data on results, selective reporting and “other bias” were taken into account.

Regarding ”other bias”, for in vitro studies, the toxic metabolites of benzene and different cell lines, dose and time of exposure were considered. For in vivo studies, the exposure conditions (e.g., route, concentration and duration of exposure) were considered. For human exposure, the sample size, the number of controls (if present), the entity of exposure and the data source were considered.

## 3. Results

Table 1, Table 2 and Table 3 summarize the key information based on the in vivo, in vitro and human studies included in this systematic review, respectively, regarding the miRNA differential expression due to benzene (or its metabolites) exposure.

### 3.1. Data from In Vitro Studies

All selected in vitro studies were carried out by exposing cell lines to the highly reactive and bioactive metabolites of benzene responsible for its toxicity, specifically 1,4-BQ and HQ. 1,4-BQ is believed to be responsible for the myelotoxicity/myeloid neoplasms observed in the bone marrow of subjects who have been exposed to high levels of benzene, even if the molecular mechanisms are not yet completely defined [13]. HQ, in addition to being an active metabolite of benzene with a potential risk for hematological disorders and hematotoxicity in humans, is a parent compound of 1,4-BQ [13,42,45].

In 2016, Chen et al. [39] exposed U937 cells for 24 h to different 1,4-BQ concentrations (0, 10, 20 and 40 μM). The induction of apoptosis was dose-dependent and after treatment (from >20 μM of 1,4-BQ), miR-133a decreased, while the pro-apoptotic genes (Caspase-9 and Caspase-3) increased. The mechanism of action of miR-133a in 1,4-BQ-exposed cells was studied using lentivirus vector transfection, resulting in miR-133a overexpression and a decrease in apoptosis. U937 cells were transfected by empty lentiviral vectors (LV-miR NC) or lentiviral vectors with miR-133a (up-LV-miR-133a). MiR-133a was upregulated by up-LV-miR-133a, and this overexpression attenuates the increase in Caspase-3 and Caspase-9.

The next year, they demonstrated the upregulation of miR-34a in the U937 cell line following the same exposure conditions (0, 10, 20 and 40 μM for 24 h). 1,4-BQ exposure causes a gradual decrease in cell viability inducing apoptosis and dose-dependent cytotoxicity, as well as the upregulation of miR-34a (from >20 μM of 1,4-BQ). The role of miR-34a was assessed using lentiviral vector transfection. Inhibiting miR-34a increased the Bcl-2 protein which reduced apoptosis induced by 1,4-BQ, demonstrating that miR-34a is involved in benzene-induced hematotoxicity through Bcl-2 [40]. Finally, the HL-60 cell line (p53 null) and the human lymphoblastoid TK6 cell line (p53 wild-type) were treated with increasing concentrations of 1,4-BQ at different doses for 24 h. MiR-222 was robustly upregulated in a dose-dependent manner in both cell lines, with significant increases starting at the lower dose in the HL-60 cells and at the 10 μM dose in the TK6 cells. Simultaneously, the DNA repair capacity became significantly reduced starting from the 10 μM dose in both cell lines [44].

Regarding HQ, to identify the miRNAs specifically involved in benzene exposure and their regulatory role in erythroid differentiation, Liang et al. [41] performed a miRNA microarray in CD34+ hematopoietic progenitor cells isolated from human umbilical cord blood after treatment with different HQ concentrations (1.0, 2.5 and 5.0 μM for 12 days).

MiRNA analysis revealed that miRNA-451a, miRNA-486-5p and miRNA-126-3p expressions were significantly lower in HQ-treated CD34+ hematopoietic progenitor cells from a concentration of 5.0 μM with respect to the untreated cells. Jiang et al. showed that the expression levels of miR-221 were significantly increased in HQ-transformed malignant cells (16HBE-T), relative to the normal controls, and that the exposure of the control cells to exosomes, which were derived from HQ-transformed malignant cells, increased miR-221 levels and promoted their proliferation [42]. Similarly, the study of Xian et al. [43] showed the inhibition of apoptosis by HQ and the role that exosomes carrying miR-221 have in the progression of benzene toxicity. The amount of miR-221 in the hydroquinone-transformed malignant cell line (16HBE-t) was significantly increased compared with the controls, and when the recipient cells ingested exosomes, derived from 16HBE-t, the miR-221 level was increased and apoptosis induced by HQ was inhibited. Lastly, in 2022, Yu et al. [45] conducted a complementary analysis of the expressions of miRNA and mRNA to identify possible molecular mechanisms between miRNAs and their target genes and the pathways of miRNA and mRNA linked to HQ-induced hematoxicity. Of the 23 miRNAs, 1108 target genes and 2304 miRNAs-mRNAs couples selected, miR-1246 and miR-224 appeared to be the main regulators of K562 cells exposed to HQ.

### 3.2. Data from Animals

Few in vivo studies have been conducted to identify the expression of miRNA after benzene exposure. Based on the selection criteria, three articles were identified where mice were exposed to either benzene alone [41,47] or in a VOC mixture [46]. To our knowledge, no articles on other animals addresses both benzene exposure and its effects on the expression of miRNA.

Liang et al. [41] evaluated the effect of the expression of miR-451a and miR-486 on the bone marrow of C57BL/6J mice, together with an examination of the periphery blood cytometry following benzene exposure by inhalation. Whole blood cell (WBC) and red blood cell (RBC) count decreased significantly, while platelet (PLT) count increased. Meanwhile, the expression of both miR-451a and miR-486 decreased significantly (about two-fold). Similarly, Wei al [47] described in the same species and strain (C57BL/6 mice) exposed to benzene by injection a different expression of the pattern of miRNA and a decreased number of peripheral white blood cells, RBC, lymphocytes and hemoglobin concentration. Furthermore, the number of various hematopoietic progenitor cells in bone marrow decreased. Five miRNAs were overexpressed, and forty-five miRNAs were downregulated. MiR-34a expression significantly changed at different levels in hematopoietic cells, confirming that it may be involved in hematopoiesis during benzene exposure, while, in contrast with the previous study, miR-451a was upregulated.

Finally, Wang et al. [46] evaluated whether lung miRNA expression in mice could be modified by an inhalation of VOCs (including benzene). Among the 1086 miRNAs detected in the mice lung tissue, 69 miRNAs were dysregulated (52 miRNA downregulated and 17 upregulated). Simultaneously, lactate dehydrogenase (LDH) and IL8 protein release significantly increased in BALF. However, in lung tissue, nitric oxide synthase (NOS) increased, whereas the content of GSH significantly decreased. Mapping the most significantly changed miRNAs to their predicted mRNA targets and their bioprocesses, molecular functions and pathways revealed the association between VOC exposure and cancer and inflammation of the lungs in mice.

### 3.3. Environmental/Occupational-Based Studies (Epidemiological Studies)

Epidemiological studies that have been carried out to evaluate the association of benzene exposures with aberrantly expressed miRNAs are summarized in Table 3. All these miRNAs were detected in blood. A recent study identified and validated abnormal miRNAs using lymphocytes from 56 benzene-poisoned workers and 53 controls. In particular, a pool of subjects was formed to investigate the relationships between benzene-induced hematotoxicity and the aberrant expression of miRNAs. From this pool, six benzene-poisoned workers and six unexposed controls (matched for age and sex) were randomly selected. The study on these subjects showed that 69 miRNAs were upregulated and 75 were downregulated. After miRNA identification and validation, 173 benzene-exposed workers (171 males and 2 females) who worked in the production of benzene and had lower exposure doses than the permissible concentration of TWA (TWA: 6 mg/m^3^) and 58 unexposed controls (35 males and 23 females) were recruited for miRNA validation, as well as genotoxicity and DNA repair capacity (DRC) studies. MiR-222 expression was upregulated among both benzene-poisoned and benzene-exposed workers, together with an inverse association with DRC [44]. In another study, 135 participants were enrolled and classified into the following three group: 54 non-exposure controls (36 men and 18 women, mean age: 38.0 ± 6.3); 54 healthy exposure subjects (36 men and 18 women, mean age: 38.1 ± 6.5); and 27 chronic benzene-poisoned patients (CBP patients) (18 men and 9 women, mean age: 38.4 ± 5.5). A total of 287 and 272 miRNAs were found to be expressed in the plasma pool of the non-exposed subjects and the healthy exposure subjects, respectively, whereas only 236 miRNAs were detected in that of CBP patients. Compared with the non-exposures and the exposures, miR-24-3p and miR-221-3p were significantly upregulated (1.99- and 2.06-fold for miR-24-3p, 2.19- and 3.93-fold for miR-221-3p, *p* < 0.01), while miR-122-5p and miR-638 were significantly downregulated (−3.45- and −2.60-fold for miR-122-5p, −1.82- and −3.20-fold for miR-638, *p* < 0.001) in the CBP patients. Compared with the non-exposures, the plasma level of miR-638 was significantly upregulated (1.38-fold, *p* < 0.01), while the plasma levels of miR-122-5p and miR-221-3p were significantly downregulated (−0.85- and −1.74-fold, *p* < 0.01) in the exposures [49]. To gain insight into the new biomarkers and molecular mechanisms of chronic benzene-poisoned, miRNA profiles and mRNA expression patterns from the peripheral blood mononuclear cells of chronic benzene-poisoned patients and health controls without benzene exposure (matched for age and sex) were performed using array platforms. Totally, six upregulated miRNAs (miR-34a, miR-205, miR-10b, let-7d, miR-185 and miR-423-5p-2) and seven downregulated miRNAs (miR-133a, miR-543, hsa-miR-130a, miR-27b, miR-223, miR-142-5p and miR-320b) were found in the chronic benzene-poisoned group compared to the healthy controls (*p* ≤ 0.05) [48]. To explore a novel potential biomarker of adverse health effects following benzene exposure, another study was conducted on 314 benzene-exposed workers and 288 control workers who were exposed to an air benzene concentration of 2.64 ± 1.60 mg/m^3^ and 0.05 ± 0.01 mg/m^3^, respectively. MiRNA-34a and benzene metabolites were determined, and positive correlations were found. Moreover, miR-34a was correlated with peripheral blood counts, alanine transaminase (ALT) and oxidative stress. In the multivariate canonical correlation analysis (CCA), miR-34a and oxidative stress indicators were significantly correlated with peripheral blood counts and ALT. In addition, miR-34a showed the greatest contribution of variables and was significantly associated with the other set of variables, particularly with WBC [40]. In another study, 100 workers were randomly recruited, of which 50 were occupationally exposed to benzene and 50 workers were exposed to negligible amounts of benzene. Each participant was required to answer a questionnaire, involving lifestyle, demographic and occupational information (e.g., gender, age, drinking history, smoking history, medication history and family history of health status). The concentrations of air benzene in benzene-exposed workers and the controls were 3.50 ± 1.60 mg/m^3^ and 0.06 ± 0.01 mg/m^3^, respectively. The MiRNAs and apoptotic pathway were examined. Caspase-9 and Caspase-3 were upregulated, while miR-133a expression decreased in benzene-exposed workers. Pearson’s correlation analysis showed that miR-133a was reversely correlated with pro-apoptotic gene Caspase-9 [39].

However, another study was carried out over the period from May 2011 to October 2014 in southern China with 97 petrol station attendants as the exposure group and 103 general residents as the control group. Plasma benzene was analyzed by using GC/MS. miR-221 in peripheral blood lymphocytes were measured by qRT-PCR. The results showed that the air concentrations of benzene were significantly higher in petrol stations than in the control sites (*p* < 0.05). The levels of benzene and miR-221 in the exposure group were both significantly higher than in the control group (*p* < 0.05) and there was a significant positive correlation between the two indexes (r = 0.851, *p* < 0.05). An association between benzene levels and ΔCt values (i.e., ΔCt = Ct_miR-221_ − Ct_U6_) for miR-221 was identified by univariate and multivariate logistic analysis (OR 0.274; 95%CI 0.117, 0.396) [50].

## 4. Discussion

Benzene is a pervasive occupational and environmental contaminant that affects human health [1,2,3]. It is well established that exposure to benzene exposure can result in hematotoxic (including immunotoxic), genotoxic and carcinogenic (i.e., leukemogenic) effects [10,11,12]. Although its metabolism is well known, the pathogenetic mechanisms through which it exerts its toxic effect, in particular on bone marrow, are not yet well defined. A new interpretation could be provided by epigenetic alterations resulting from exposure to benzene (or its metabolites), including low doses [16,18,21]. In this review, particular attention was paid to miRNAs as important physiological or pathological regulators of cellular molecular mechanisms resulting from in vitro, in vivo and human epidemiological studies.

The results of the in vitro studies demonstrate and reinforce the use of this model, together with the bioinformatics approach, to understand the molecular mechanisms of benzene toxicity. In this perspective, it is important to determine the role of miRNAs in the activation process to understand which miRNAs are involved and activated, as well as how, to study them in the following animal model.

The toxic effects of benzene on the human body are mainly mediated by its bioactive metabolites, among which HQs and BQs are the major representative. The study of Yu et al. 2022 [45] reinforces the use of an in vitro model of HQ exposure and bioinformatic approaches to advance our knowledge on the molecular mechanisms of benzene hematotoxicity at the RNA level. In particular, they show how miR-1246 and miR-224 may be potential biomarkers for the assessment of benzene hemotoxicity. Both miRNAs have a crucial role in hemotoxicity: miRNA-1246 is upregulated to activate the expression of C6orf211 and C19orf10 to promote tumor progression [51], and miR-224 is impaired in the marrow of pediatric AML patients [52]. Liang et al. 2018 [41] showed how the downregulation of specific miRNAs played a role in promoting erythroid cell differentiation. In the studies conducted by Jiang et al. 2019 [42] and Xian et al. 2021 [43], the human bronchial cell line 16HBE was transformed into a malignant one by exposure to increasing concentrations of HQ, which expressed a high level of miR-221. The exosomes derived from these cells can transmit miR-221 to normal recipient cells to promote cell viability and proliferation, confirming the important role played by this miRNA in the control of the molecular mechanisms underlying the toxicity of benzene. All these results provide new insights into the mechanism of benzene-induced erythroid suppression and into the identification of potential biomarkers for the evaluation of benzene-induced erythroid hemotoxicity by strengthening the use of the in vitro model.

In vitro studies, in addition to clarifying molecular mechanisms, may also be used to support evidence from epidemiological studies by which, in addition to quantifying the incidence of pathological conditions, biomarkers of exposure/effect can be identified. This is the case in Chen’s studies [39,40] where the downregulation of miR-133a and the upregulation of miR-34a in benzene-exposed workers was confirmed in the U937 cell line in vitro. Both in vitro studies confirmed the molecular mechanisms underlying the above miRNAs’ deregulation due to benzene exposure. In the same way, in the study of Wang et al. [44], benzene-poisoned workers and the matched controls showed the deregulation of several miRNAs. Among these, miRNA-222 was confirmed to be robustly upregulated in a dose-dependent manner in two cell lines. The high expression of miRNA-222 led to the failed activation of p53, abnormal DNA repair capacity and serious biological consequences. This scenario can be considered a novel and key process for benzene-induced hematotoxicity and leukemogenesis [44].

Other studies have only considered an epidemiological approach with no in vitro model. The study of Liu et al. [49] explored the potential effects of occupational benzene exposure on plasma miRNA expression considering chronic benzene exposure patients, healthy benzene-exposed individuals and non-benzene-exposed individuals. They identified four miRNAs (miR-24-3p, miR-122-5p, miR-221-3p and miR-638) as potential biomarkers that could also provide new insights into the pathogenesis of chronic benzene exposure. These miRNAs have been reported to regulate genes that could influence the adverse effects of benzene exposures or the progression of chronic poisoning. miR-24-3p can regulate several important target genes which might be mainly involved in oxidative damage regulation. MiR-122-5p might play a cell-cycle arrest role in DNA-damaging repair where miR-221 may be considered a generalized inhibitor of myelopoiesis by inhibiting molecules involved in myeloid development. MiR-638 repress the BRCA1 protein level and its loss alters the DNA double-strand-break repair [49]. The expression of some miRNAs has been also related to both benzene metabolites and biomarkers of nucleic acid damage [53]. Other studies have found the implication of different miRNAs. Bai et al. [48] identified a set of miRNAs whose aberrant expression might be considered a potential biomarker of chronic benzene poisoning. With a different approach, Hu et al. [50] showed that benzene exposure may be related to elevated miRNA-221 expression in human lymphocytes. MiR-221 is altered in a number of malignancies and significantly deregulated in the blood cells of patients with leukemia [54,55].

Many articles have studied the effects of in vivo benzene exposure, but very few take the role of miRNA into consideration, and only do so in mice. The study of Liang et al. [41] showed that the suppression of miRNA-451a and miRNA-486-5p might be tied to the benzene-induced adverse impact on bone marrow erythroblastic cells. In the same way, benzene induces alteration in hematopoiesis and hematopoiesis-associated miRNAs [47]. Most of these miRNAs have been shown to be dysregulated in various hematological malignancies [56,57,58,59] and in the regulation of hematopoiesis, such as in the differentiation and proliferation of hematopoietic stem and progenitor cells [60,61,62]. Finally, the inhalation of VOCs, including benzene which alters miRNA patterns, potentially leads to the initiation of cancer and inflammatory diseases [49]. These in vivo studies cannot provide comprehensive conclusions due to the limited number of articles. However, they show how the expression of various miRNAs is altered in exposed mice, although these miRNAs are different from those found in epidemiological studies. These studies also show hamatotoxic effects that, according to published data, are more severe in males than in females [12].

Even if the profile of miRNA expression following benzene exposure varies from cultured cells to animal models and to adults, it is clear that their alteration is associated with chemical exposure, as has already been reported by other authors [63,64].

In light of various studies, how should miRNAs be considered: biomarkers of “dose” or “effect”? A biomarker of “dose” indicates the dose of xenobiotic absorbed by an organism and represents the contribution of all possible sources, but not the existence of an adverse effect. If the biomarker of “dose” is associated with a biomarker of “effect”, the relationship between exposure and biological modification (observed response) can be established [65,66]. Indeed, a biomarker of “effect” responds to biological variations that may affect the biochemical, molecular or cellular components, and it may be associated with the onset of disease, providing useful information that can reduce harmful effects. This relationship yields the identification of inter and intra individual variations over time in relation to physiological conditions and changes in the body prior to disease onset. Given the significance of the “dose “and “effect” biomarkers, miRNAs seem to approach and follow the behavior of the biomarker of effect, proving useful as diagnostic and prognostic indicators [67,68]. Recent studies have showed that human miRNAs are involved in the pathogenesis of many diseases as participants in processes such as cell proliferation, signal transduction, inflammation and immune response [69]. Moreover, miRNAs are stable and can be very specific to different types of tissues and cellular types of these tissues and can become a therapeutic tool [70].

However, there are also limitations in considering miRNAs as biomarkers of effect. Firstly, by their nature, miRNAs are non-specific biomarkers [31,69], and the interpretation of their alteration becomes even more complex when co-exposure, the most common condition in work and environmental settings, occurs [48,50,53]. This does not change the fact that studying their profile (rather than a single miRNA) may provide valuable insights into molecular mechanisms by which the xenobiotic exerts its biological activity [29,63,71]. For benzene, this is crucial because its mechanisms of action are not yet fully understood. In this contest, to better evaluate the cause–effect relationship, it might be useful to integrate the information obtained from the variation in the profiles of miRNAs with other biomarkers of effect (e.g., biomarkers of nucleic acids damage) and/or biomarkers of exposure [56].

Another factor to consider is that miRNAs may change over time and during a lifespan depending on different physiological and life conditions (e.g., lifestyle, aging, pathology, exposure to risk factors, etc.) [69,72]. Consequently, a unique and direct miRNA signature that reflects exclusively benzene exposure is difficult to find.

Despite this, by integrating the expression data of miRNA and mRNA, it was shown that the different expressions of miRNAs are implicated in the molecular pathway of benzene toxicity. This opens new paths of prevention against chronic benzene poisoning through miRNA pharmacological interventions.

Another aspect that should be studied is the long-term effects of benzene exposure [31]. It will be interesting to understand if miRNA deregulation has transient character. Long-term longitudinal studies would allow us to estimate the contribution of miRNAs to disease development [73]. Similarly, early or prenatal exposure to toxic substances could alter the expression of microRNA and these alterations could lead to adverse outcomes later in life [63,64]. In this respect, future generations should be monitored for the inheritance of alterations in miRNA expression. Furthermore, it would be important to evaluate the effects of benzene exposure on disease developing progressively during adulthood and the role played by miRNAs.

## 5. Conclusions

MiRNAs are biomarkers also able to offer great potential in the occupational medicine field, and they could also be used to understand the mechanism of benzene carcinogenicity. In fact, they are present in all human biological fluids, and they are easily measured by means of commercial kits. Therefore, the use of miRNAs may help to identify the effects of a specific exposure with benefits for disease prevention and health promotion.

This study confirms the potential role of miRNAs in the biological response induced by environmental/occupational carcinogens. It also underlines the importance of identifying miRNA signatures specific to benzene exposures, which, once validated, will help to improve workplace safety conditions.

## Figures and Tables

**Figure 1 ijerph-20-01920-f001:**
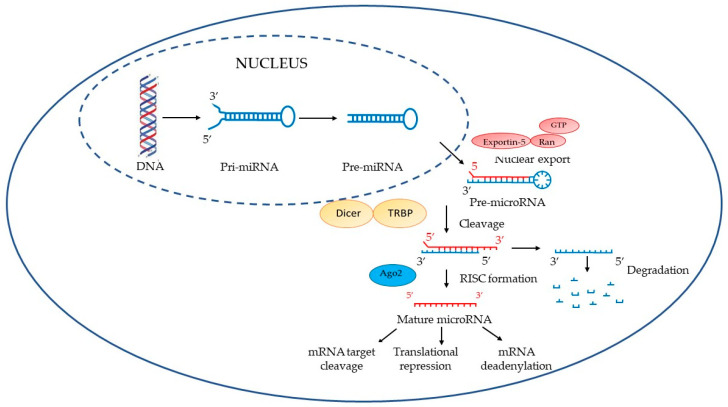
Biogenesis of miRNA. Adapted from Winter et al., 2009 [30] and Vrijens et al., 2015 [31].

**Figure 2 ijerph-20-01920-f002:**
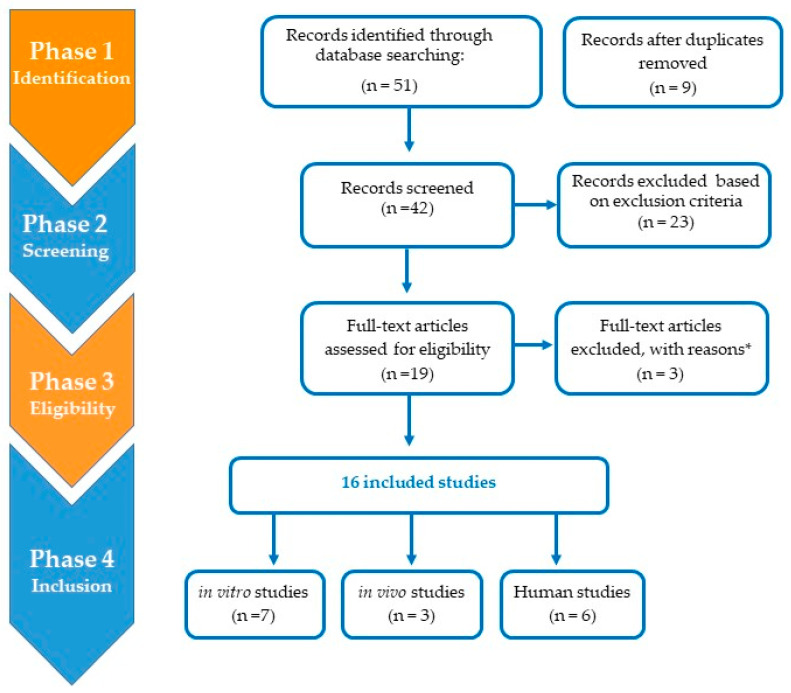
Flowchart of the selection process for articles. * When characterization of benzene exposure is not fully described.

**Table 1 ijerph-20-01920-t001:** The main outcomes from in vitro studies.

Reference	Cells Types	Exposure Conditions	miRNAs	Main Endpoints
Chen Y et al. 2016 [39]	U937 ^1^ cell line	1,4-BQ ^2^ (0, 10, 20 e 40 μM) for 24 h	miRNA-133a	1,4-BQ ^2^-induced apoptotic cell death is dose-dependent. MiR-133a decreased while pro-apoptotic genes (Caspase-9 and Caspase-3) increased after 1,4-BQ ^2^ treatment. miR-133a is inversely correlated with Caspase-9.
Chen Y et al. 2017 [40]	U937 ^1^ cell line	1,4-BQ ^2^ (0, 10, 20 e 40 μM) for 24 h	miRNA-34a	1,4-BQ ^2^-induced apoptosis upregulates miR-34a in the U937 ^1^ cell line; inhibition of miR-34a elevated Bcl-2 ^3^ protein and reduced apoptosis. miR-34a is involved in benzene-induced hematotoxicity through Bcl-2.
Liang B et al. 2018 [41]	CD34+ HPCs ^4^ cells and K562 ^5^ cell line	HQ ^6^ at different concentrations for 24 h	miRNA-451a miRNA-486-5p miRNA-126-3p	miRNA-451a and miRNA-486-5p were upregulated during erythroid differentiation in both CD34 + HPCs ^4^ and K562 ^5^ cells. Benzene exposure leads to the suppression of miR-451a and miR-486 expression.
Jiang R et al. 2020 [42]	16HBE ^7^ cell line	HQ ^6^ (0, 5, 10, 20, 40, 80 g/mL) for 24 h	miRNA-221	Normal cells (16HBE ^7^) can be transformed into malignant ones by HQ ^6^. MiR-221 has an elevated expression level in HQ ^6^-transformed malignant cells.
Xian HY et al. 2021 [43]	16HBE ^7^ cell line and HQ ^4^-transformed 16HBE ^7^ (16HBE-t) cells	HQ ^6^ (0, 5, 10, 20, 40, 80 g/mL) for 24 h	miRNA-221	The amount of miRNA-221 in 16HBE ^7^-t was increased compared with controls. When recipient cells ingested exosomes derived from 16HBE ^7^-t, miR-221 was increased and apoptosis induced by HQ ^6^ was inhibited.
Wang TS et al. 2021 [44]	human HL-60 ^8^ (p53 null) and TK6 ^9^ (p53 wild-type) cells	1,4-BQ ^2^ (2.5, 5, 10 and 20 μM) for 24 h	miRNA-222	miRNA-222 was robustly upregulated in a dose-dependent manner in both cell lines, with significant increases starting at the 2.5 μM dose in the HL-60 ^8^ cells and at the 10 μM dose in the TK6 ^9^ cells.
Yu CH et al. 2022 [45]	K562 cells ^5^	40 μM HQ ^6^ for 72 h	The regulatory network of miRNAs includes 23 miRNAs.	MiR-1246 and miR-224 had the potential to be major regulators in HQ ^6^-exposed K562 cells based on the miRNAs-mRNAs network.

^1^ U937 cell line: histiocytic lymphoma cell line; ^2^ BQ: benzoquinone; ^3^ Bcl-2: B-cell lymphoma 2; ^4^ CD34+ HPCs cells: umbilical cord blood cells; ^5^ K562 cell line: chronic myelogenous leukemia cell line; ^6^ HQ: hydroquinone; ^7^ 16HBE cell line: human; ^8^ HL-60 cell line: human promyeoloblasts; ^9^ TK6 cell line: human lymphoblast cell line.

**Table 2 ijerph-20-01920-t002:** The main outcomes from in vivo studies.

Reference	Species and Strain	Exposure Conditions	miRNAs	Main Endpoints
Wang F et al. 2014 [46]	44 male Kunming mice	VOCs ^1^ exposure: benzene: 3, 3–5–10 mg/m^3^ 2 weeks at 2 h/day.	69 miRNAs measured in lung were Significantly differentially expressed in VOCs’ ^1^ exposed samples	VOCs ^1^ exposure potentially alters signaling pathways associated with cancer, chemokine signaling, Wnt signaling, neuroactive ligand–receptor interaction and cell adhesion molecules.
Wei H et al. 2015 [47]	C57BL/6 mice	Benzene exposure by injection: 150 mg/kg benzene for 4 weeks.	5 miRNAs were overexpressed, and 45 miRNAs were downregulated in bone marrow	5 miRNAs were overexpressed and 45 miRNAs were downregulated in the benzene exposure group, where also decreased both the number of cells in peripheral blood and hematopoietic progenitor cells in the bone marrow.
Liang B et al. 2018 [41]	C57BL/6J mice (32 mal and 32 female)	Benzene exposure by inhalation: 0, 1, 5, 25 ppm for 14–28 days (6 h/day and 6 days/week).	miRNA-451a miRNA-486-5p in bone marrow	The expression of miRNA-451a or miRNA-486-5p was negatively correlated with the concentration of benzene inhalation on erythroid toxicity of C57BL/6J mice.

^1^ VOCs: volatile organic compounds.

**Table 3 ijerph-20-01920-t003:** The main outcomes from human epidemiological studies.

Reference	Population in Study	Exposure Conditions	miRNAs	Main Endpoints
Bai W et al. 2014 [48]	4 patients of chronic benzene poisoning three benzene-exposed workers 3 health controls without benzene exposure	Airborne benzene concentration between health controls without benzene exposure and chronic benzene poisoning group is 0.06 ± 0.01 mg/m^3^ and 6.68 ± 2.28 mg/m^3^, respectively.	6 upregulated miRNAs 7 downregulated miRNAs	Compared to health controls, hsa-miR-34a, hsa-miR-205, hsa-miR-10b, hsa-let-7d, hsa-miR-185 and hsa-miR-423-5p-2 were upregulated miRNAs in chronic benzene poisoning group, while hsa-miR-133a, hsa-miR-543, hsa-miR-130a, hsa-miR-27b, hsa-miR-223, hsa-miR-142-5p and hsa-miR-320b in chronic benzene poisoning group are downregulated miRNAs compared with controls.
Liu Y. et al. 2016 [49]	27 chronic benzene poisoning patients- low blood counts 54 healthy benzene-exposed individuals 54 non-exposed individuals	Subjects for microarray analysis: exposure intensity (mg/m^3^): 13.9 ± 6.9 Subjects for validation analysis: exposure intensity (mg/m^3^): 14.2 ± 8.1.	miRNA-24-3p miRNA-221-3p miRNA-122-5p miRNA-638	miRNA-24-3p and miRNA-221-3p were significantly upregulated miRNA-122-5p and miRNA-638 were significantly downregulated in the CBP patients with respect to exposed and non-exposed. miRNA-638 was significantly upregulated and miRNA-122-5p and miRNA-221-3p were significantly downregulated in the exposed with respect to non-exposed.
Chen Y et al. 2016 [39]	50 benzene-exposed workers 50 controls subjects	The concentrations of air benzene in benzene-exposed workers and controls were 3.50 ± 1.60 mg/m^3^ and 0.06 ± 0.01 mg/m^3^, respectively.	miRNA-133a	miRNA-133a expression decreased in benzene-exposed workers and Caspase-9 and Caspase-3 were simultaneously upregulated.
Hu D et al. 2016 [50]	97 petrol station attendants as exposed group 103 general residents as control group	Air concentration of benzene (μg/m^3^) for exposed group: 72.99 ± 18.81 Air concentration of benzene (μg/m^3^) for control group: 7.47 ± 1.15	miRNA-221	The levels of benzene and miRNA-221 in exposure group were both significantly higher than in control group (*p* < 0.05) and there was a significant positive correlation between the two indexes (r = 0.851, *p* < 0.05).
Chen Y et al. 2017 [40]	314 benzene-exposed workers 288 non-exposed workers.	Air benzene concentration for exposed workers: 2.64 ± 1.60 mg/m^3^. Air benzene concentration for exposed workers: 0.05 ± 0.01 mg/m^3^	miRNA-34a	miRNA-34a expression was elevated in benzene-exposed workers and was correlated with the airborne benzene concentration, S-PMA ^1^ and t, t-MA ^2^.
Wang TS et al. 2021 [44]	73 current benzene-exposed workers 8 non-exposed controls.	The maximum detected TWA ^3^ was 1.51 mg/m^3^ and the mean of TWA ^3^ was 0.37 mg/m^3^	miRNA-222	miRNA-222 expression was upregulated in benzene-exposed workers, together with inverse association with DRC ^4^.

^1^ S-PMA: S-phenylmercapturic acid; ^2^ t, t-MA: trans, trans-muconic acid; ^3^ TWA: time-weighted average; ^4^ DRC: DNA repair capacity.

## Data Availability

No new data were created or analyzed in this study. Data sharing is not applicable to this article.

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
