# Peer review of "Benzene Exposure and MicroRNAs Expression: In Vitro, In Vivo and Human Findings"

_ijerph, 2023, doi:10.3390/ijerph20031920_

Round 1

Reviewer 1 Report

The manuscript “Benzene Exposure and MicroRNAs Expression: in Vitro, in Vivo and Human Findings” by Mozzoni et al. focuses on an innovative area of ​​study with great potential for future research, the expression of miRNAs, applied to a relevant field in human health, the exposure to benzene. It is generally well written, but some corrections and additions to the text are needed, with major changes required in the discussion section and clarifications in the methodology applied for article selection. Namely:

Abstract

Line 17 – include “changes” after “miRNA expression”

The abstract should include the major findings obtained from this review, as mentioned in the title, and not only general information about the relevance of using miRNA studies.

Introduction

Line 51 - delete spaces in “2008/50 / EC”

Line 80 – misses “are” before “considered”

Line 82 – write in full ROS before using the abbreviation for the first time, and use only ROS in line 84

Line 87-89 – Correct the sentence: The epigenetic alterations comprise several mechanisms, including changes in DNA methylation, in histone tails, and in microRNAs (miRNAs) expression.

Line 90 – the global genomic hypomethylation of (reference 17 is in what model? Specify to understand the difference from reference 18 that also refers global hypomethylation.

Line 92-94 - very confusing sentence. Acetylation and methylation are in the histone tails, not in gene promoters. Rewrite to clarify how histone tail modifications interfere with gene promoters.

Line 109 – Correct the sentence: MiRNAs are usually made of 18–25 nucleotides, and are extremely specific in tissue-specific

Line 111 – delete “region”, since it is already written in “UTR”-

Line 111 - Translation is from mRNAs to proteins, and the sentence is confusing. Rewrite: …inhibit their translation, regulating several physiological processes. Delete “as they are involved in the regulating target genes at the post-transcriptional level” since it is already written some lines before.

Figure 1 – this figure is a copy of figure 1 of Winter, J., Jung, S., Keller, S. et al. Many roads to maturity: microRNA biogenesis pathways and their regulation. Nat Cell Biol 11, 228–234 (2009). https://doi.org/10.1038/ncb0309-228. Refer that this figure is adapted from the previous one and be cautious if you are not violating authorship rights.

Line 129 – keep the same nomenclature all manuscript: microRNAs or miRNAs

Line 133-134 – The sentence “Their profile could become a useful tool to predict a toxicant-target that interacts and defines individual susceptibility to toxicants” is not clear. how can a profile that is supposed to serve as a biomarker with universal application define individual susceptibility? Is this not defined by other genetic differences, such as SNPs? Clarify.

Materials and methods

How many articles were initially retrieved using the keyword strings you defined? Specify their number by keyword string. I suppose that some (for example, "benzene and exposure") would result in a high number of articles not related to miRNAs. How was the selection made? Did you choose the ones that had miRNA data by reading the abstracts or using other informatic filtering methods?

Figure 2 - the number of records after duplicate removal it is not 9, its 43. you can merge the two boxes "records after duplicated removed” and “records screened” since they are the same thing. Also, detail in the text why 3 articles were excluded, if they are not foreseen in the exclusion criteria.

Line 173 – substitute “pathways” by “routes”

Why did you not consider the dose and time of exposure as a bias?

Results

Line 187 – write “differential” before “expression”

Table 1.b - It would be useful to add the tissues from where the miRNAs were extracted, since they are tissue-specific.

Table 1.c – Refer somewhere if all studies were performed in blood samples. In the study of Hu D et al 2016, delete “The air concentrations of benzene were significantly higher in petrol stations than in control sites (p<0.05)” since this information is not relevant to this article.

Line 211 – Correct “They demonstrated that apoptotic cell death induced is dose dependent…” to “they demonstrated that the induction of apoptosis is dose-dependent…”

Detail at what concentrations did miR-133a decreased while pro-apoptotic genes (Caspase-9 and Caspase-212 3) increased. All?

Line 123- Correct the sentence: “The mechanism of miR-133a in 1,4-BQ-exposure…” to “the mechanism of action of miR-133a in 1,4-BQ exposed cells…”

Line 214 - it is not given sufficient information about the transfection to the reader understand why the results are contrary to the effects previously reported

Again, detail the concentration that up-regulated miR-34a in the U937 cell line.

Line 225- in the study of Liang et al, Is there any difference between the results obtained at different concentrations? At what concentration were the miRNAs lower? And what was the exposure time?

Line 231 – write “the” before “expression levels”

Line 235 – delete “induced” and write “carrying” between exosomes and miR-221

Line 241 – “routes and arrays”???

Line 280 – correct “shows” to “showed”

Line 340- 342 - correct “P” to “p

Discussion

All discussion must be reviewed in order to delete the text that only describes results (for instance, lines 362-367; 388-395; 403-406; 418-427;430-446). All discussion is full of duplicated information already provided in the results section. If some of these data is not included in that section, that information must be merged into a single text. To discuss the results from these studies (which of course is the objective), simply cite the study in question or describe its major findings in brief, but not exhaustively and repeatedly as it is.

Not only the benefits, but also the limitations of using miRNAs as biomarkers of effect must be discussed. For instance, the specificity of the differentially expressed miRNAs to benzene exposure, since many of these miRNAs (miR-221 e miRNA-222, for instance) were also found differentially expressed in many other situations were cell viability and differentiation are affected. Thus, they may not be considered specific biomarkers of benzene exposure and would only have that interpretation in workers exclusively exposed to benzene. Also, the question of miRNA changes over time.  Although the last paragraph focuses this question, it is not only their long-term changes that matter. To be a biomarker of effect, the miRNAs profile must stay the same. The time of exposure of the studies here reported is not adressed in the discussion, but after one week exposure, the miRNA profile may not be the same than after 24h. Other limitations should also be identified based on the studies here considered, and here discussed.

Line 395 - delete “an”

Line 475-477 – nonsense sentence. Clarify.

Author Response

REFEREE 1

Comments and Suggestions for Authors

The manuscript “Benzene Exposure and MicroRNAs Expression: in Vitro, in Vivo and Human Findings” by Mozzoni et al. focuses on an innovative area of ​​study with great potential for future research, the expression of miRNAs, applied to a relevant field in human health, the exposure to benzene. It is generally well written, but some corrections and additions to the text are needed, with major changes required in the discussion section and clarifications in the methodology applied for article selection. Namely:

Abstract

Line 17 – include “changes” after “miRNA expression”

  1. This word has been added.

The abstract should include the major findings obtained from this review, as mentioned in the title, and not only general information about the relevance of using miRNA studies.

  1. The abstract has been modified based on the reviewer’s suggestion.

Introduction

Line 51 - delete spaces in “2008/50 / EC”

  1. The spaces have been deleted.

Line 80 – misses “are” before “considered”

  1. The required change has been made.

Line 82 – write in full ROS before using the abbreviation for the first time, and use only ROS in line 84

  1. The required change has been made.

Line 87-89 – Correct the sentence: The epigenetic alterations comprise several mechanisms, including changes in DNA methylation, in histone tails, and in microRNAs (miRNAs) expression.

  1. The required change has been made.

Line 90 – the global genomic hypomethylation of (reference 17 is in what model? Specify to understand the difference from reference 18 that also refers global hypomethylation.

  1. The required change has been made. A new reference has been added.

Line 92-94 - very confusing sentence. Acetylation and methylation are in the histone tails, not in gene promoters. Rewrite to clarify how histone tail modifications interfere with gene promoters.

  1. This sentence has been rewritten.

Line 109 – Correct the sentence: MiRNAs are usually made of 18–25 nucleotides, and are extremely specific in tissue-specific

  1. The required change has been made.

Line 111 – delete “region”, since it is already written in “UTR”-

  1. The required change has been made.

Line 111 - Translation is from mRNAs to proteins, and the sentence is confusing. Rewrite: …inhibit their translation, regulating several physiological processes. Delete “as they are involved in the regulating target genes at the post-transcriptional level” since it is already written some lines before.

  1. The required change has been made.

Figure 1 – this figure is a copy of figure 1 of Winter, J., Jung, S., Keller, S. et al. Many roads to maturity: microRNA biogenesis pathways and their regulation. Nat Cell Biol 11, 228–234 (2009). https://doi.org/10.1038/ncb0309-228. Refer that this figure is adapted from the previous one and be cautious if you are not violating authorship rights.

  1. Figure has been manually typed using "power point" based on several examples from literature. We decided to change the colours and layout so that it is clear the lack of copyright. However, we agree that the caption should include some references to the articles we used to compose the new figure.

Line 129 – keep the same nomenclature all manuscript: microRNAs or miRNAs

  1. The required change has been made.

Line 133-134 – The sentence “Their profile could become a useful tool to predict a toxicant-target that interacts and defines individual susceptibility to toxicants” is not clear. how can a profile that is supposed to serve as a biomarker with universal application define individual susceptibility? Is this not defined by other genetic differences, such as SNPs? Clarify.

  1. This sentence has been rewritten.

Materials and methods

How many articles were initially retrieved using the keyword strings you defined? Specify their number by keyword string. I suppose that some (for example, "benzene and exposure") would result in a high number of articles not related to miRNAs. How was the selection made? Did you choose the ones that had miRNA data by reading the abstracts or using other informatic filtering methods?

  1. We thank the reviewer for the thorough analysis and we apologize for this misunderstanding. In fact, selection was carried out using as first keywords “benzene and microRNA” and, as second keywords: “exposure”, “biomonitoring”, “human”, “in vitro”, “and “in vivo”. This choice prevented a large number of irrelevant articles from being selected, as the reviewer also pointed out. Lastly, the final choice was made by first reading the abstract and then the full article.

All these aspects have been highlighted in paragraph 2.1. and 2.2.

Figure 2 - the number of records after duplicate removal it is not 9, its 43. you can merge the two boxes "records after duplicated removed” and “records screened” since they are the same thing. Also, detail in the text why 3 articles were excluded, if they are not foreseen in the exclusion criteria.

  1. We apologize for the mistake. Figure 2 has been modified following the suggestion of the two reviewers. Details related to the box “Full-text articles excluded, with reasons” have been added to the figure caption:

“*elements for characterisation of benzene exposure are not described”.

Line 173 – substitute “pathways” by “routes”

  1. The required change has been made.

Why did you not consider the dose and time of exposure as a bias?

  1. We agree with the reviewer’s suggestion and the test has been modified accordingly.

Results

Line 187 – write “differential” before “expression”

  1. The required change has been made.

Table 1.b - It would be useful to add the tissues from where the miRNAs were extracted, since they are tissue-specific.

  1. The required change has been made.

Table 1.c – Refer somewhere if all studies were performed in blood samples PAOLA. In the study of Hu D et al 2016, delete “The air concentrations of benzene were significantly higher in petrol stations than in control sites (p<0.05)” since this information is not relevant to this article.

  1. The required changes have been made.

Line 211 – Correct “They demonstrated that apoptotic cell death induced is dose dependent…” to “they demonstrated that the induction of apoptosis is dose-dependent…”

  1. The required change has been made.

Detail at what concentrations did miR-133a decreased while pro-apoptotic genes (Caspase-9 and Caspase-212 3) increased. All?

  1. These elements have been included in the paper.

Line 123-213 Correct the sentence: “The mechanism of miR-133a in 1,4-BQ-exposure…” to “the mechanism of action of miR-133a in 1,4-BQ exposed cells…”

  1. The required change has been made.

Line 214 - it is not given sufficient information about the transfection to the reader understand why the results are contrary to the effects previously reported

  1. Required details have been included in the paper.

Again, detail the concentration that up-regulated miR-34a in the U937 cell line.

  1. Required details have been included in the paper.

Line 225- in the study of Liang et al, Is there any difference between the results obtained at different concentrations? At what concentration were the miRNAs lower? And what was the exposure time?

  1. Required details have been included in the paper.

Line 231 – write “the” before “expression levels”

  1. The required change has been made.

Line 235 – delete “induced” and write “carrying” between exosomes and miR-221

  1. The required change has been made.

Line 241 – “routes and arrays”???

  1. These sentences has been more clearly expressed.

Line 280 – correct “shows” to “showed”

  1. The required change has been made.

Line 340- 342 - correct “P” to “p

  1. The required change has been made.

Discussion

All discussion must be reviewed in order to delete the text that only describes results (for instance, lines 362-367; 388-395; 403-406; 418-427;430-446). All discussion is full of duplicated information already provided in the results section. If some of these data is not included in that section, that information must be merged into a single text. To discuss the results from these studies (which of course is the objective), simply cite the study in question or describe its major findings in brief, but not exhaustively and repeatedly as it is.

  1. The discussion has been rewritten and reorganized based on input from both reviewers.

Not only the benefits, but also the limitations of using miRNAs as biomarkers of effect must be discussed. For instance, the specificity of the differentially expressed miRNAs to benzene exposure, since many of these miRNAs (miR-221 e miRNA-222, for instance) were also found differentially expressed in many other situations were cell viability and differentiation are affected. Thus, they may not be considered specific biomarkers of benzene exposure and would only have that interpretation in workers exclusively exposed to benzene. Also, the question of miRNA changes over time.  Although the last paragraph focuses this question, it is not only their long-term changes that matter. To be a biomarker of effect, the miRNAs profile must stay the same. The time of exposure of the studies here reported is not adressed in the discussion, but after one week exposure, the miRNA profile may not be the same than after 24h. Other limitations should also be identified based on the studies here considered, and here discussed.

  1. We thank the reviewer for these important suggestions which definitely make the review more comprehensive and objective.

Line 395 - delete “an”

  1. The required change has been made.

Line 475-477 – nonsense sentence. Clarify.

  1. This concept has been discussed more clearly elsewhere, so we decided to delete the sentence because it does not add value to the discussion.

Finally, in order to respond to reviewers’ questions, we have decided to add new references:

Saliminejad, K.; Khorram Khorshid, H.R.; Soleymani Fard, S.; Ghaffari, S.H. An overview of microRNAs: Biology, functions, therapeutics, and analysis methods. J Cell Physiol. 2019, 234(5), 5451-5465.

Winter, J.; Jung, J.; Keller, S.; Keller, S.; Gregory, R.I.; Diederichs, S. Many roads to maturity: microRNA biogenesis pathways and their regulation. Nat Cell Biol  2009, 11, 228–234

Bersimbaev, R.; Pulliero , A.; Bulgakova, O.; Asia, K.; Aripova, A.; Izzotti, A. Radon Biomonitoring and microRNA in Lung Cancer. Int J Mol Sci 2020, 21 (6), 2154.

Hu, J.; Ma, H.; Zhang, W.; Yu, Z.; Sheng, G.; Fu, J. Effects of benzene and its metabolites on global DNA methylation in human normal hepatic L02 cells. Environ Toxicol 2014, 29 (1), 108-116

Reviewer 2 Report

The manuscript reviews the published studies on benzene-induced microRNA expression/profile changes reported in benzene exposed humans and in in vitro and in vivo studies. It also aims to compare the reported results from these three types of studies in order to better understand the molecular mechanisms behind benzene-induced toxicity, as well as evaluating whether miroRNA might serve as a future biomarker of benzene exposure and/or benzene-induced effects.

Overall, the paper is well written and easy to follow, and it gives a good background for why the review is needed. Nevertheless, the manuscript would benefit from a language editing, including checking spelling, grammar and sentence structure. I have some comments below that might further improve the manuscript:

Introduction

1. The two first paragraphs should be shortened, and be limited to what is relevant for the review.

2. Line 86-87: The study (ref. 16) is said to be a recent study. The study is from 2017 (5 years old), and cannot be said to be new within this discipline.

Material and methods

3. The flow chart given in figure 2 is transparent. However, I suggest to move all boxes indicating the papers excluded on the right side (“Records after duplicates removed (n=9)”), and only keep the papers included in the downstream flow.

Results

4. The authors must ensure that the units given for the reviewed papers are correct. At least two of them are wrong;

-          Line 51: 5 mg/m3 should be 5 µg/m3

-          Table 1 c (page 9): In the column “Exposure conditions” an air concentration of 72.99 and 7.47 mg/m3 is given for the exposed group and control group, respectively, in the study by Hu D et al. 2016 [47]. The correct unit is µg/m3.

5. Reference 48 (Sisto et al. 2019) should be deleted both from table 1 and from the discussion since benzene exposure is a contaminant in the painting operation, and not the main exposure of interest in the study. Also, environmental exposure is not given nor in the referred study or in the table, in which ethylbenzene is given in the latter (not relevant for benzene in this context),

6. The last column in tables 1 a, b and c should be removed from the table as it seems to be concluded by the authors of the reviewed studies, and not as an evaluation done by the authors of the present manuscript. The information in this column can be moved to the discussion section.

Discussion

7. The paper would have benefited from structuring the discussion according to the aims given in the introduction.

Author Response

REFEREE 2

Comments and Suggestions for Authors

The manuscript reviews the published studies on benzene-induced microRNA expression/profile changes reported in benzene exposed humans and in in vitro and in vivo studies. It also aims to compare the reported results from these three types of studies in order to better understand the molecular mechanisms behind benzene-induced toxicity, as well as evaluating whether miroRNA might serve as a future biomarker of benzene exposure and/or benzene-induced effects.

Overall, the paper is well written and easy to follow, and it gives a good background for why the review is needed. Nevertheless, the manuscript would benefit from a language editing, including checking spelling, grammar and sentence structure. I have some comments below that might further improve the manuscript:

Introduction

  1. The two first paragraphs should be shortened, and be limited to what is relevant for the review.
  2. According to the reviewer’s suggestion, the two first paragraphs have been condensed in the next sentence.

“Benzene is a ubiquitous environmental pollutant produced by natural (e.g. natural gas, petroleum, combustion processes, etc. ) and anthropogenic processes (e.g. industrial and vehicular emissions, chemical processes, cigarette smoke, etc.).”

  1. Line 86-87: The study (ref. 16) is said to be a recent study. The study is from 2017 (5 years old), and cannot be said to be new within this discipline.
  2. We agree and therefore the words “Most recently” have been deleted.

Material and methods

  1. The flow chart given in figure 2 is transparent. However, I suggest to move all boxes indicating the papers excluded on the right side (“Records after duplicates removed (n=9)”), and only keep the papers included in the downstream flow.
  2. We agree and Figure 2 has been modified according to the suggestion of the two reviewers.

Results

  1. The authors must ensure that the units given for the reviewed papers are correct. At least two of them are wrong;

-          Line 51: 5 mg/m3 should be 5 µg/m3

-          Table 1 c (page 9): In the column “Exposure conditions” an air concentration of 72.99 and 7.47 mg/m3 is given for the exposed group and control group, respectively, in the study by Hu D et al. 2016 [47]. The correct unit is µg/m3.

  1. We thank the reviewer for the thorough analysis. We apologize for these errors probably due to the final font setting.

  1. Reference 48 (Sisto et al. 2019) should be deleted both from table 1 and from the discussion since benzene exposure is a contaminant in the painting operation, and not the main exposure of interest in the study. Also, environmental exposure is not given nor in the referred study or in the table, in which ethylbenzene is given in the latter (not relevant for benzene in this context),
  2. We agree and the reference Sisto et al has been removed from the table c. As a result, Figure 2, Result and Discussion have been modified.

  1. The last column in tables 1 a, b and c should be removed from the table as it seems to be concluded by the authors of the reviewed studies, and not as an evaluation done by the authors of the present manuscript. The information in this column can be moved to the discussion section.
  2. According to reviewer’s suggestion, the last column has been removed in all the tables and the information have been inserted in discussion.

Discussion

  1. The paper would have benefited from structuring the discussion according to the aims given in the introduction.
  2. The discussion has been rewritten and reorganized based on input from both reviewers.

Finally, in order to respond to reviewers’ questions, we have decided to add new references:

Saliminejad, K.; Khorram Khorshid, H.R.; Soleymani Fard, S.; Ghaffari, S.H. An overview of microRNAs: Biology, functions, therapeutics, and analysis methods. J Cell Physiol. 2019, 234(5), 5451-5465.

Winter, J.; Jung, J.; Keller, S.; Keller, S.; Gregory, R.I.; Diederichs, S. Many roads to maturity: microRNA biogenesis pathways and their regulation. Nat Cell Biol  2009, 11, 228–234

Bersimbaev, R.; Pulliero , A.; Bulgakova, O.; Asia, K.; Aripova, A.; Izzotti, A. Radon Biomonitoring and microRNA in Lung Cancer. Int J Mol Sci 2020, 21 (6), 2154.

Hu, J.; Ma, H.; Zhang, W.; Yu, Z.; Sheng, G.; Fu, J. Effects of benzene and its metabolites on global DNA methylation in human normal hepatic L02 cells. Environ Toxicol 2014, 29 (1), 108-116

Round 2

Reviewer 1 Report

Just correct:

Line 205 – misses “to” before benzene

Line 284 - Patways to pathways

Line 433- “both cell lines”: Not seeing the table of results at the same time, it is difficult to understand the transition between the exposed workers and “both cell lines”. It would be better to replace “Among these, miRNA-222 was robustly up-regulated in a dose-depend manner in both cell lines” with “Among these, miRNA-222 was confirmed to be robustly up-regulated in a dose-depend manner in two cell lines.”

Line 464 – write in full HSPCs

Line 466 – delete “that regulate gene expression” as it is completely unnecessary.

Line 502 - mirRNA to miRNA

Line 504 - RNA to miRNA

Author Response

REFEREE 1

Comments and Suggestions for Authors

Just correct:

Line 205 – misses “to” before benzene

  1. The required change has been made (in green).

Line 284 - Patways to pathways

  1. The required change has been made (in green).

Line 433- “both cell lines”: Not seeing the table of results at the same time, it is difficult to understand the transition between the exposed workers and “both cell lines”. It would be better to replace “Among these, miRNA-222 was robustly up-regulated in a dose-depend manner in both cell lines” with “Among these, miRNA-222 was confirmed to be robustly up-regulated in a dose-depend manner in two cell lines.”

  1. The required change has been made (in green).

Line 464 – write in full HSPCs

  1. The required change has been made (in green).

Line 466 – delete “that regulate gene expression” as it is completely unnecessary.

  1. We agree and it has been deleted.

Line 502 - mirRNA to miRNA

  1. The required change has been made (in green).

Line 504 - RNA to miRNA

  1. The required change has been made (in green).

Reviewer 2 Report

Thank you for considering my comments and suggestions. I have no other comments. 

Author Response

REFEREE 2

Comments and Suggestions for Authors

hank you for considering my comments and suggestions. I have no other comments. 

We have really appreciated your suggestions, which certainly make the review more complete, clear and objective.